# Perceptions and behaviors of healthcare providers towards rehabilitation support to children with severe malaria-related disability in Ethiopia: A qualitative descriptive study using the Theoretical Domains Framework

Eshetu Haileselassie Engeda[1]*, Heather M. Aldersey[2], Colleen M. Davison[3], Kassahun Alemu Gelaye[4], Nora Fayed[2]

1 Department of Pediatric and Child Health Nursing, School of Nursing, College of Medicine and Health Sciences, University of Gondar, Gondar, Ethiopia, 2 Queen's University School of Rehabilitation Therapy, Kingston, Ontario, Canada, 3 Department of Public Health Sciences, Queen's University Kingston, Kingston, Ontario, Canada, 4 Department of Epidemiology and Biostatistics, Institute of Public Health, College of Medicine and Health Sciences, University of Gondar, Gondar, Ethiopia

* 18ehe@queensu.ca

## Abstract

### Introduction

Severe malaria often results in childhood disability. The prevalence of disability related to severe malaria is significant and is estimated to affect up to 53% of severe malaria survivors. In contrast, information is sparse about how healthcare providers in Africa think about or provide rehabilitation support in acute and post-acute phases respectively. Understanding the perceptions and behaviors of healthcare providers treating malaria could help inform malaria-related disability research, policy, and practice, aimed at the providers themselves. This study explored the perceptions and behaviors of healthcare providers towards rehabilitation for children with severe malaria-related disability. The Theoretical Domains Framework was used to describe the findings relative to wider literature on health provider behavior change.

### Methods

A qualitative descriptive approach was used to interview thirteen healthcare providers recruited purposively based on their clinical professions, roles, and settings. Data were analyzed using directed content analysis. We decided on the most prominent theoretical domains considering the frequency of specific perceptions and behaviors across the participants, the frequency of perceptions and behaviors in each domain, and evidence of strong perceptions and behaviors.

**Data Availability Statement:** All relevant data are within the paper and its Supporting Information files.

**Funding:** This work was supported by the Mastercard Foundation Scholars Program, which covered personal and material costs as part of a larger research project. Only EE received the grant (grant number not applicable). However, the organization had no role in designing or conducting the study, including data collection, management, analysis, interpretation of the findings, and manuscript writing, review, and approval.

**Competing interests:** The authors have declared that no competing interests exist.

## Results

Nine out of fourteen theoretical domains were identified. These domains were: Beliefs about consequences, environmental context and resources, goals, knowledge, skills, optimism, reinforcement, social influences, and social or professional role and identity. Healthcare providers' beliefs about their roles in screening for disability or referring to rehabilitation were less positive.

## Conclusions

The findings of this study suggest the need for interventions to support healthcare providers in acute phases (prevention and control of severe malaria) and post-acute phases (disability screening, referral, and rehabilitation care). Recommended interventions should focus on developing clinical guidelines, training clinicians, addressing institutional factors, and modifying external social influences such as socio-cultural factors.

## Introduction

Severe malaria is a laboratory-confirmed malaria characterized by one or more clinical symptoms or laboratory findings, including impaired consciousness, respiratory distress, multiple convulsions, prostration, abnormal bleeding, jaundice, hypoglycemia, acidosis, hyperlactatemia, renal impairment, hyperparasitemia, cerebral malaria, and severe malarial anemia [1]. In line with the World Health Organization's standard definition of disability, any form of impairment (e.g., attention deficits), activity limitation (e.g., behavioral problems), and participation restriction (e.g., negative peer attitudes) that occurred because of this health condition was considered as severe malaria-related disability [2,3]. Evidence shows that severe malaria is associated with long-term health consequences in children. It has been linked with neurocognitive [4–8], mental health [9,10], behavioral [11,12], and developmental [7,13] sequalae. The prevalence of disability is significant and is estimated to affect up to 53% of severe malaria survivors in some African studies [7,14]. The most studied severe malaria-related components of disability affecting child functioning were identified in a recent scoping review of the African malaria literature using the International Classification of Functioning, Disability, and Health (ICF) [15]. Additional evidence was also obtained from an Ethiopian qualitative study on severe malaria-related disability among children from caregivers' perspective that uncovered the psychosocial components of disability in addition to biological impairments [8]. Thus, children who experience disability associated with severe malaria need to obtain the necessary health, rehabilitation, and educational services to maximize their long-term functioning. Knowledge of whether or how the current healthcare delivery system addresses severe malaria-related disability in the African and Ethiopian contexts is unknown.

In the global context, in general, and in the Ethiopian context, in particular, the existing practice of post-malaria clinical management focuses on correcting immediate complications such as hypoglycemia, renal failure, and anemia [16]. During acute phases of the disease health workers focus on a medical approach that includes but not limited to blood transfusion (for severe anemia), correction of blood glucose level, seizure management, correction of fluid and electrolyte imbalance, and dialysis or renal replacement therapy [16–18]. Known rehabilitation interventions for children with severe malaria-related components of disability have focused on neurocognitive impairment using cognitive training software [19–21]. Description or

availability of models of practice, systems for early screening, and referrals of children with severe malaria-related disability were not found. Rehabilitation interventions that have been effective for children with disability, such as inclusive schools [22–24], and community-based rehabilitation services [25–27] are often available throughout Africa, yet it is unknown whether children with severe malaria-related components of disability are linked to these services.

The declining child mortality because of the availability of better antimalaria medicines (such as artesunate) [28–31] and the significant number of malaria-exposed children in sub-Saharan Africa [32,33] indicate that healthcare systems need to recognize the significant impacts and requirements to provide responses to prevent or address severe malaria-related disability. As with other forms of childhood neurodisabilities, studying the perceptions and behaviors of the healthcare providers towards the prevention of severe malaria-related disability and provision of rehabilitation support when components of disability occur are crucial [34].

A paucity of research on rehabilitation interventions for children with severe malaria-related disability can be balanced with evidence on childhood disability from other infectious causes. For example, acute bacterial meningitis (the most common infectious cause of childhood neurocognitive disability), cytomegalovirus infections, and syphilis are global research priorities, affecting populations in lower-and middle-income countries [35]. The management of the long-term impacts of pediatric meningitis includes rehabilitation, ideally comprised of three significant components: i) awareness of the problem by health professionals, and accessibility and availability of facilities to address the components of disability; ii) community-based rehabilitation services aimed at the health of children with components of disability (e.g., improving access to rehabilitation services), and education (e.g., increasing school attendances); and iii) social participation (e.g., improving relationships with peers and removing environmental barriers) [36]. The extent to which healthcare providers are utilizing similar approaches in the case of children with severe malaria-related disability are unknown along with their perception of components of disability associated with severe malaria.

The Theoretical Domains Framework (TDF) helps to explore healthcare providers' perceptions and behaviors of desirable healthcare practices in order to improve health systems through positive behavior change by clinicians [37]. Although it has been applied in various healthcare contexts [38–41] and is a useful way to conceptualize care providers' views and behaviors, it has not been used to understand whether, or how disability and rehabilitation are addressed in current malaria health practices. In this study, we explored the perceptions and behaviors of healthcare providers towards providing rehabilitation support to children with severe malaria-related disability using the TDF.

## Materials and methods

### Target perceptions and behaviors

In this study, the target perceptions and behaviors under investigation were in favor of preventing severe malaria-related disability and providing rehabilitation support when severe malaria-related disability occurs. Prevention of severe malaria-related disability is characterized by managing severe malaria well enough that the chance of disability (impairment, activity limitation, or participation restriction) is less or none. Preventive activities include, but are not limited to, early identification and prompt treatment of severe malaria (ideally, it is the management of acute cases within 24 hours) [16], and immediate referral of complex malaria cases. The provision of rehabilitation health services help maximize "functioning" among children with severe malaria-related disability. Here, "functioning" is defined as the overall well-being of a child in terms of physical health (body structures and body functions), activities (e.g., activities of daily life), and social health (e.g., participation in various social events) [2].

The rehabilitation interventions can be provided at a health facility level or at the community level to improve the functioning and health of children with severe malaria-related disability, education, inclusion, and social participation [42,43]. Such a service also includes early identification and providing rehabilitation support for children with severe malaria-related disability by establishing screening mechanisms (prevention of further disability).

## Study design

A qualitative descriptive design was used. This approach is a design of choice when a phenomenon is described straightforwardly. Data presentation involves a descriptive summary of the relevant contents of data logically [44]. We explored the phenomenon of interest in its natural state to the extent possible within the study context without being pre-occupied by pre-existing theories of methods. The qualitative descriptive design is the least "theoretical" of all qualitative designs [44–46]. The design also allows for more freedom in terms of commitment to a theoretical framework as compared to methodologies such as phenomenology and grounded theory, and it provides freedom to begin with (or not to begin with) a theory of the phenomenon of interest [45]. In line with this notion, we used the TDF in designing and conducting our study.

## Study setting

The study was conducted in purposively selected healthcare facilities found in malaria-endemic areas of Northwest Ethiopia. We also considered the level of the healthcare facilities according to Ethiopia's healthcare delivery system (primary, secondary, and tertiary).

## Context

In Ethiopia, the first contact place for children with malaria (including severe cases) is a health post, which is part of the Primary Health Care Unit (PHCU), and it is the closest and easily accessible health facility (a few minutes' drive if transportation is available or less than two hours on foot). A PHCU comprises five health posts and one health center. A health post in Ethiopia is staffed with two female health extension workers; their role is mainly preventive, but they also manage non-complicated cases of malaria and refer severe cases to a health center [47–49]. The difference between a health post and a health center is in staffing and facilities, where the latter is staffed with several nurses, health officers, laboratory technicians, and pharmacy professionals. Moreover, health centers are equipped with essential laboratory services such as a blood film, urine analysis, blood cell counts, and hemoglobin tests, whereas health posts have only rapid diagnostic tests for malaria. Health centers treat non-complicated malaria and some severe cases; however, they still need to refer severe malaria cases to the next level, a district (primary) hospital, which is staffed with general practitioners (in addition to nurses and other health professionals).

District (primary) hospitals are equipped with better treatment and diagnostic facilities compared to health centers, having organ function tests, various imaging tests (mainly X-rays and ultrasounds), and better anti-malaria medicines (prescribed by physicians). The health posts, the health centers, and primary hospitals together form the primary level of care. If children with severe malaria need further diagnosis or treatment, primary hospitals refer them to general hospitals (secondary level care), staffed with specialists such as pediatricians. Most severe malaria cases can be managed at general hospitals. However, if children have problems that need sub-specialists such as a pediatric neurologist or need advanced diagnostic facilities such as MRI, the general hospitals refer them to a specialized hospital (tertiary level care) [50].

## Participants

Participants were recruited from PHCUs, district hospitals, and specialized hospitals intending to capture the viewpoints of healthcare providers working at various levels of the country's healthcare delivery system. Therefore, health extension workers, nurses, health officers, general practitioners, and pediatricians were included in the study. These experts were chosen because they were most likely to be directly involved in treating and caring for children with severe malaria.

## Sampling

Thirteen healthcare providers were recruited using the purposive sampling technique based on the type of profession, relevant clinical experience, and the level of healthcare facility. Ten initial interviews for preliminary analysis were conducted followed by three additional interviews, according to the recommendations of Francis and colleagues for theory-informed qualitative studies [51]. Additionally, the concept of information power was used to determine when to stop recruiting participants, considering the study's aim, sample specificity, quality of dialogue, and analysis strategy [52].

## Data collection

Face-to-face interviews were conducted with a TDF-based interview guide (S1 Table). EE conducted all interviews in the local language (Amharic) using the guide between April 2021 and August 2021. The interviews took place at the healthcare providers' offices at times that were convenient for them (e.g., when there were no clients or patients). Prior to the actual study, a pilot interview with one healthcare provider was conducted in the research setting, intending to evaluate the interview guide. As a result, some questions were re-written for clarity. Moreover, the interview guide was refined throughout the process based on subsequent interviews [53]. The average length of the interviews was 50 minutes, and all of them were audiotaped using two recording devices. As all the recordings were audible to transcription and gave sufficient data for analysis, repeated interviews were unnecessary, and no participants refused to participate. EE took field notes following each interview, intending to record issues related to the context, non-verbal expressions of participants, and feelings, as recommended by Phillipi and Lauderdale (2018) [54]. All interview records were transcribed verbatim by research assistants and cross-checked with the audio records.

## Data analysis

We imported the transcripts into NVivo 12 Plus software to facilitate the data analysis. The purpose of the software was to assist the researchers in organizing, visualizing, storing, and reporting their data; otherwise, the entire analysis was a cognitive process. To minimize loss of meaning during translation [55,56], we analyzed the data using Amharic (the local language) until we generated preliminary themes. After that, the codes and associated data were translated into English. Data were analyzed deductively using the TDF. In addition, EE developed a coding guide (S2 Table), according to the recommendations highlighted in a published TDF guide [37], which facilitated the coding process and minimized coding inconsistencies between multiple coders.

We used the directed content analysis technique and followed different phases involved in the approach [57]. First, EE developed a data analysis matrix, defined the theoretical domains based on the study's context, and developed a coding guide according to the TDF. At this phase, EE read and re-read the interview transcripts several times to "immerse in the data"

[58]. Second, EE and his colleague independently pilot coded two interview transcripts. At this stage, by discussing and resolving the challenges of applying the TDF, the researchers ensured coding consistency and agreed on a coding strategy that further improved the coding guide. The researchers also discussed the initial findings of the two transcripts before coding the remaining data. Furthermore, a senior researcher (NF) reviewed these initial findings and gave valuable recommendations. Third, the remaining data were coded by EE and NF using the updated coding guide, and the reliability of coding was checked first by proportion and then by Kappa statistics. The other team members were involved in a discussion to resolve a few codes that the two coders disagreed on, which helped finalize the coding process. This stage involved reading and re-reading transcripts, identifying meaning units (interesting items of data that help form themes and sub-themes), writing summarized meaning units, preliminary coding, clustering codes into categories, coding the contents into themes and subthemes, and mapping the themes or subthemes into the appropriate TDF domains. Table 1 presents examples of the steps involved in the directed content analysis. Finally, the researchers decided on the most relevant TDF domains based on three criteria: the frequency of specific perceptions and behaviors across the participants, the frequency of perceptions and behaviors in each domain, and evidence of strong perceptions and behaviors directly influencing the perceptions and behaviors of healthcare providers [37].

## Trustworthiness

To achieve "prolonged engagement," [58] EE spent several months in the study areas, carrying out activities such as building relationships, arranging appointments and interview places, conducting a pilot interview, and conducting the main interviews. As he had worked in similar settings over ten years, EE was familiar with the Ethiopian healthcare delivery system at each level, and that helped him to quickly engage in the setting and build trust. To minimize preconceptions that could negatively affect the process and outcomes of the study, EE was regularly monitoring and documenting his relationships with the participants and the way he reacted to the participants' views (reflexivity) [59]. The reflexive journaling continued throughout the study, including data analysis to minimize bias. The participants were aware of the purpose of doing the research. To ensure credibility, there were periodical debriefing

**Table 1. An example of the steps involved in the directed content analysis.**

| Meaning unit | Summarized meaning unit | Preliminary code | Group of codes | Subtheme/specific belief | TDF domain |
|---|---|---|---|---|---|
| "Our duty becomes more manageable when materials are complete, but this is not always the case. For example, we refer a mother with a febrile child when an RDT [Random Diagnostic Test for Malaria] is unavailable" (R309). | Materials are incomplete | Lacking materials | Lacking materials | Scarcity of required resources | Environmental context and resources |
| "I think the long-term consequences need professionals specially trained for that purpose. However, unfortunately, we now lack such practitioners who can adequately assess and manage these issues" (R303). | We lack professionals specialized in this area | Lacking specialized professionals | Lacking specialized professionals | Absence of relevant guideline | Environmental context and resources |
| "No, there has not been a protocol for long-term issues until now. There is, however, one for treating acute malaria, about first-line medications, second-line medications, and so on" (R301). | There is no treatment protocol for the long-term complications | Lacking treatment protocol | Lacking relevant protocol | | |
| "To be honest, the guideline we are using focuses solely on treating acute malaria. So, it is all about dealing with acute malaria at the health post, health center, and hospital levels" (R305). | The existing guideline focuses only on treating acute malaria | Lacking relevant guideline | Lacking relevant guideline | | |

sessions among the research team members, especially with a senior researcher (NF) [60]. These sessions helped improve the whole study process including subsequent interviews, data analysis, and report writing.

We kept contextual, methodological, and analytical documentation to facilitate an audit (decision) trail [61]. The contextual documentation included an explicit description of the study setting and participants, while the methodological documentation encompassed a detailed description of the methodologies, methods, and procedures used. The analytical documentation included a detailed description of analytical and theoretical insights obtained from the data. This documentation also helped us to maintain a "thick description" of our findings. At two separate stages of the study, we completed member checks [62]. The first happened during the actual interview when EE reflected on his understanding of the participants' statements for most responses, and they often verified that they were on the same page. The second came after the analysis was finished. EE provided a copy of the preliminary findings to a group of three participants (a health extension worker, a nurse, and a physician) to read it independently. EE then asked for their feedback and enabled a group discussion. The participants did not generate new ideas, nor do they oppose or amend the existing ones; instead, they agreed with the aggregated findings and felt that their experiences were valued.

## Ethical considerations

Ethical approval was granted by Queen's University in Ontario, Canada, and the University of Gondar in Ethiopia. The University of Gondar wrote a letter of support to the concerned administrative authorities in the study areas in Amharic (local language). Prior to each interview, we got both oral and written informed consent. To ensure confidentiality, we removed all names and other identifiers from transcripts and records, presented the findings anonymously, and only the research team had access to the data.

## Results

### Participant characteristics

Thirteen healthcare providers participated in the study, all of whom had direct clinical experience treating children with malaria. The majority (53.8%) came from Primary Health Care Units (PHCU) and had a median of six years of relevant clinical experience (IQR = 4–10). Furthermore, most of the participants (61.5%) were men. Table 2 summarizes the sociodemographic characteristics of participants included in the study.

### Healthcare providers' perceptions and behaviors

Nine relevant theoretical domains (out of fourteen) were identified that described the perceptions and behaviors of healthcare providers in preventing severe malaria-related disability and providing rehabilitation support when the components of disability occur. Table 3 presents the theoretical domains (and subthemes) that influence the healthcare providers' perceptions and behaviors in preventing severe malaria-related disability and providing rehabilitation support when the components of disability occur. We also delineated the proportion of codes that addressed the prevention of severe malaria-related disability, indicating acute care of severe malaria instead of rehabilitation support and care (S3 Table).

### Theoretical domain one: Knowledge

"Knowledge" is characterized by the healthcare providers' expression of their existing knowledge about severe malaria-related disability, comprehensive approaches to identify the

**Table 2. Sociodemographic characteristics of the study participants (n = 13).**

| Participant code | Sex | Profession | Clinical experience in years | Health facility |
|---|---|---|---|---|
| R301 | M | Pediatrician | 8 | Specialized hospital |
| R302 | M | Health officer | 6 | PHCU (Health center) |
| R303 | M | BSc nurse | 4 | District hospital |
| R304 | F | BSc nurse | 10 | PHCU (Health center) |
| R305 | M | Physician general practitioner | 2 | District hospital |
| R306 | M | Health officer | 27 | PHCU (Health center) |
| R307 | F | BSc nurse | 4 | PHCU (Health center) |
| R308 | M | Physician general practitioner | 6 | District hospital |
| R309 | F | Health extension worker | 14 | PHCU (Health post) |
| R310 | M | Pediatrician | 6 | Specialized hospital |
| R311 | F | Health extension worker | 5 | PHCU (Health post) |
| R312 | F | Health extension worker | 10 | PHCU (Health post) |
| R313 | M | Pediatrician | 5 | Specialized hospital |

PHCU = Primary Health Care Unit.

components of disability (e.g., understanding of severe malaria-related disability from the biopsychosocial perspective), and prevention of severe malaria-related disability and providing rehabilitation support. In the subtheme "We do not know much about it," most participants described that they barely know about severe malaria-related disability.

"It is a little bit challenging to talk about this issue because it needs a follow-up or further investigation as we do not know much about it. To know such a case, children who have survived severe malaria should seek health care in our facility. . . . but unfortunately, I cannot exactly tell what kind of health problems such a child might have developed" (R302).

**Table 3. Theoretical domains and subthemes.**

| Theoretical domain | Subtheme |
|---|---|
| 1. Knowledge | "We do not know much about it" |
| 2. Skills | "We need the training to fill our gaps" |
| 3. Beliefs about consequences | Beliefs about severity of illness |
| | Believes about disease outcomes |
| 4. Goals | Planning to consider such problems in the future |
| | Planning to work in the preventive aspects |
| 5. Optimism | "I hope things will improve in the future" |
| 6. Reinforcement | Rewarding conditions |
| | Discouraging conditions |
| 7. Environmental context and resources | Absence of relevant guideline |
| | Scarcity of relevant resources |
| | Systems and processes affecting treatment |
| 8. Social influences | Patient beliefs |
| | Socio-cultural influences |
| 9. Social or professional role and identity | Clinical practice inconsistencies |
| | Beliefs about the scope of practice |
| | "We work together, and the team spirit is good" |

Others highlighted that severe malaria could have long-term health impacts in surviving children recalling their clinical education; however, they did not see it from the disability point of view and had not encountered such cases.

"The long-term effects may require more research; I did not experience any health issues like these. I agree that malaria can have consequences, but I have not seen such cases. However, when you consider the scientific evidence about malaria's pathophysiology, there is no way it can be free of long-term complications" (R305).

## Theoretical domain two: Skills

"Skills" is characterized by the healthcare providers' expression of the required competencies to manage severe malaria-related disability. In the subtheme, "We need the training to fill our gaps," most participants expressed that they need skills-based training to address severe malaria-related disability as they lack specialized training in this aspect.

"We do not have specialized training to assess and manage these complications; we are managing some related problems using our clinical experiences, so we need additional training to fill our gaps" (R302).

Some also highlighted their specific need for training, stating the type of long-term consequences and their management approaches should be given attention.

"It would be fantastic to fill our skills gaps through training, emphasizing the types of consequences and management measures. I would be pleased to follow those approaches and deal with the problems" (R312).

## Theoretical domain three: Beliefs about consequences

The domain "Beliefs about consequences" is characterized by healthcare providers' descriptions of illness severity, long-term consequences, or outcomes of addressing severe malaria-related disability. Participants mainly identified two substantial issues in managing severe malaria-related disability: the characteristics of severe malaria and its outcomes. For instance, in the subtheme "Beliefs about the severity of illness," the healthcare providers explored various severity-related concerns such as the defining characteristics of severity, how severely ill children presented to their health facility, and the need for urgent management for such children. Some participants emphasized that considering several parameters to determine severity has paramount importance to manage the problems properly.

"I think we need to consider various factors while determining whether or not it is severe malaria. Most children cannot stand on their own and stumble when they attempt to do so. . . . they are generally accompanied by a high-grade fever and low blood pressure. When we send a blood sample to a lab, the results usually show a high [malaria] parasite load. In addition, most children develop anemia. . . . they also lose concentration . . ." (R305).

The healthcare providers also acknowledged that children with severe malaria such as cerebral malaria must obtain urgent treatment and care within 24 hours to prevent adverse outcomes such as death and long-term complications. The participants frequently raised pertinent matters in the subtheme "beliefs about disease outcomes," such as the adverse impact

of delayed treatment, the various long-term consequences that could impair child functioning, and the need for a follow-up to prevent these consequences. Some highlighted the immediate consequences and indicated the possibility of long-term consequences.

". . . if they arrive early, it is simple to cure: most recover quickly. Some children may suffer hypoglycemia or anemia, but both are treatable if caught early. On the other hand, the younger ones may die if they contract cerebral malaria. If they survive the death, I believe they need follow-up" (R301).

The participants also acknowledged that such problems were not commonly addressed as part of their routine professional activity; however, they still believe that the consequences could exist.

"Although we have not paid much attention to these issues, I believe the disease [severe malaria] has several long-term effects. Some suffer splenomegaly and anemia, and if they are frequently attacked, they become vulnerable to other diseases; some may also have developmental issues" (R313).

## Theoretical domain four: Goals

Goals include any plans, commitments, or priorities set by the healthcare providers to take actions related to the prevention of severe malaria-related disability and provision of rehabilitation support. In this domain, the healthcare providers mainly describe two aspects of ideas: addressing severe malaria-related disability in the future and planning to work in preventive activities. In the subtheme "Planning to consider such problems in the future," the participants stated that they would address severe malaria-related disability in the future, as they have not been practicing it, and the issue is relatively new to them.

"This issue is new, so it is better to give it due attention in the future. Although many people do not know much about the long-term impacts of severe malaria, we will work on all the required activities to identify cases. We will also try to manage the problems as much as we can" (R307).

In the subtheme "Planning to work on preventive aspects," the healthcare providers express that they have a plan to work in the preventive aspects of severe malaria-related disability for the future. Some associate this concept with the indispensable role of preventing illness before severe consequences occur with unnecessary healthcare costs.

"My goal is to prevent children from being ill. So, I would like to focus on the preventative aspects. Why should people be made to suffer? Apart from the illness, there are medical costs, human resource issues, and ups and downs. So, I think it is better to focus on prevention" (R302).

## Theoretical domain five: Optimism

Optimism is characterized by the ambitions of the healthcare providers that are expressed in a way that they will successfully prevent severe malaria-related disability in the future and provide rehabilitation support when components of disability occur, hoping that the necessary conditions including resources, guidelines, and policies will be fulfilled in the future. In the

subtheme "I hope things will improve in the future," the healthcare providers expressed their hope that addressing severe malaria-related disability would be implemented in the future, indicating that there are related initiatives (including policy) put into action by various stakeholders including the Federal Ministry of Health.

"I am hopeful that policies and strategies will be designed and implemented, with non-governmental organizations assisting in the process. Despite any shortcomings on the part of the implementers [health professionals], I am confident that things will be fine in the future" (R303).

Some participants described their hope considering the emerging evidence regarding severe malaria-related components of disability, expressing their hope that this evidence might influence policymakers to some extent and help design intervention approaches.

"Things, in my opinion, will improve in the future. Researchers like you will present evidence to policymakers, who then might design policies and strategies to address the health problems associated with severe malaria's long-term consequences. For example, a guideline for dealing with these issues might be available" (R310).

### Theoretical domain six: Reinforcement

"Reinforcement" is characterized by the healthcare providers' expression of any financial or non-financial incentives that positively influence the healthcare providers' behavior on preventing severe malaria-related disability and providing rehabilitation support. It also includes negative mechanisms such as punishment or other disappointing circumstances that adversely affect their professional practice. In the subtheme "Rewarding conditions," some participants state that favorable circumstances motivate them to carry out their responsibilities. However, for some participants, the driving force is children's health.

"I really become delighted when a child recovers from his illness. You will be thrilled if you see the child playing or see him anywhere. That is highly rewarding" (R309).

Others believed that motivations come from the inside of oneself and from accomplishing the required task that ultimately helps the quick recovery of sick children.

". . . the first is internal motivation; the second is that performing your duties will make you happy. I mean, it feels pleasant on the inside. . . . but above all, it inspires when the child regains his health" (R312).

In the subtheme "Discouraging conditions," some healthcare providers expressed that there are discouraging conditions that demotivate them so that they would become reluctant for new initiatives. Some relate these issues to a lack of incentives.

"We have been complaining about one thing for a long time but do not get ears that hear: the issue of incentives. To be honest, when compared to other sectors, the health sector is significantly impacted regarding this issue" (R305).

Some healthcare providers associate these issues with external situations, especially with parents' actions when they take their severely ill children to traditional healers.

"You know what? When children develop complications, their parents usually take them to holy water and bring them to us when life-threatening conditions happen. So, my only option would be to refer them to a hospital, which made me feel desperate" (R306).

## Theoretical domain seven: Environmental context and resources

This domain is characterized by the healthcare providers' description of any institutional factors that influence their behavior in preventing severe malaria-related disability and providing rehabilitation support, such as specific management guidelines, human resources, physical resources, time, working environment, and so on. The most frequently highlighted topics in this aspect are the lack of relevant guidelines, scarcity of required resources, and systems and processes affecting the prevention of severe malaria-related disability and provision of rehabilitation support. In the subtheme "Absence of relevant guideline," the participants expressed the lack of management guidelines in various ways. Some clearly described the unavailability of a guideline for long-term consequences and highlighted the nature of the existing acute treatment guideline.

".. . no, there has not been a protocol for long-term issues until now. There is, however, one for treating acute malaria, about first-line medications, second-line medications, and so on" (R301).

The health care providers described the scarcity of several types of resources that could help address severe malaria-related disability in the subtheme "Scarcity of required resources." Although the type and number of resources specified vary depending on the level of the healthcare facility, the problem exists everywhere. Participants from specialized hospitals discussed the lack of both human and material resources.

"Pediatricians alone might not be adequate. It would be helpful if there were clinicians who specialized in neurology. That is just one example from the perspective of human resources. On the other side, as some diagnostic devices such as CT-scan and MRI are absent, it is challenging to decide on the types and stages of the problems. So, generally, essential investigations and diagnostic modalities are incomplete" (R301).

In the subtheme "Systems and processes affecting treatment," the healthcare providers most frequently described the inconsistency and ineffectiveness of the patient referral system. They specifically mentioned the lack of a feedback system that ultimately results in a lack of follow-up mechanisms for children who survived severe malaria. The participants elaborated that they could not trace children who had been treated in higher health facilities after they completed their treatment and returned to the community.

"The referral system is weak. When we transfer patients to a hospital, we frequently experience a disconnect. The professionals there frequently do not see the referral paper and, as a result, do not provide us with feedback, and that prevents us from following up on the health conditions of the children who have survived" (R302).

On the other side of the spectrum, many participants described that the focus of the health facilities could be diverted when new pandemics or epidemics emerge, and these factors negatively influence the budget allocated for malaria-related activities. The issues raised in relation to the healthcare facilities' systems and processes are not limited to the referral system and

priority setting. Some participants also described problems related to patient data recording and the challenges in obtaining previous patient records.

"... for example, when a child I treated for malaria before a week or a month comes again for a health condition, the previous history may not be available as his chart might disappear or challenging to find it. So, we may not know the link between his current health problem and his previous malaria attack" (R313).

## Theoretical domain eight: Social influences

"Social Influences" are characterized by the behaviors of healthcare providers in seeking opinions from other people about the prevention of severe malaria-related disability and provision of rehabilitation support. External influences from colleagues, patients, or the views of colleagues and other people are also the other features of this domain. For example, in the subtheme "Patients' beliefs," the healthcare providers stated that patient perceptions or beliefs influence the management of severe malaria-related disability. In addition, some participants described this issue in terms of the reluctant nature of many parents for referrals.

"When we refer some malaria survivors to higher-level health facilities for further investigation or better management, parents frequently refuse and insist that we do everything we can. . . . but, of course, life in this district is difficult due to financial constraints" (R302).

In the subtheme "Socio-cultural influences," the healthcare providers expressed those social and cultural factors that influence the management of severe malaria-related disability. In this aspect, the most frequently described factor is the tendency of parents to associate some of the manifestations of severe malaria complications with "evil eyes" or "evil spirits" and, as a result, take the children to traditional healing areas such as holy water services.

"The rural community, in particular, waits a long time to seek medical help for their children. Parents believe it is "evil eyes" and take their children to holy water or traditional healers when severe cases show seizure and epilepsy-like signs. So, they often come after the damage is severe" (R301).

Some healthcare providers also described that many parents interfere in the management of severe malaria, especially when it comes to parenteral medications.

"It is a deep-rooted belief! When they just brought the child, especially if he comes while conscious, they do not allow us to provide him injectable medications. They would disagree with parenteral routes unless they first visited the holy water treatment" (R306).

Others expressed that those parents usually come to health facilities when the children's conditions become more complicated.

"Even if a child is shivering with an acute malarial attack, there is a strong desire to turn to traditional healers, holy water, or witches. When the child's condition worsens, it is common to blame "Satan" or an "evil spirit" for the problems. As a result, parents come to us when things get out of hand" (R313).

### Theoretical domain nine: Social or professional role and identity

"Social or professional role and identity" is characterized by healthcare providers' description of expectations, professional roles, boundaries, responsibilities, job descriptions, and so on that include comparing of their roles and responsibilities with colleagues or other professionals in terms of preventing severe malaria-related disability and providing rehabilitation support. In this regard, the most frequently highlighted issues were non-uniformity in clinical practice, the scope of practice, and team spirit. In the subtheme "Clinical practice inconsistencies among professionals," the healthcare providers described the existence of inconsistencies among professionals that influence the prevention of severe malaria-related disability and provision of rehabilitation support. Some of them described it from the perspective of identifying severity.

> "When health practitioners suspect severe malaria, they order appropriate tests, such as blood sugar checks to rule out hypoglycemia, which is one of the severe presentations; however, such practices are not uniform. In some circumstances, some professionals may not even evaluate severity. In this regard, there are inconsistencies" (R302).

In the subtheme "Beliefs about the scope of practice," the healthcare providers' stated their professional roles, boundaries, and scope of practice in terms of preventing severe malaria-related disability and providing rehabilitation support. Some believe that it is included within their professional role, and others do not believe so.

> "I believe it has something to do with my job. So, if I come into a problem like this, I will talk to colleagues or malaria experts at district health offices about supporting the children who are impacted" (R304), whereas another participant put it, "I do not think this is within my scope of practice" (R309).

Finally, in the subtheme "We work together, and the team spirit is good," the healthcare providers' expressed their interaction or relationship with colleagues and other healthcare teams, including the team spirit, in terms of preventing severe malaria-related disability. Almost all expressions indicate the presence of good team spirit and collaboration among professionals within a facility.

> "Physicians, nurses, and other health professionals are present as this is a hospital. I work with them, and we have a great team spirit. Nurses carry out whatever is expected of them in the absence of physicians, which is also true in the absence of nurses. . . . we constantly collaborate and function as a team" (R305).

## Discussion

The healthcare providers in this study emphasized a belief that they should prevent severe malaria-related disability with better acute care management as opposed to post-acute screening, monitoring, or rehabilitation referral. Although the healthcare providers discussed long-term management in some domains in-depth, their major discussion points were a lack of information, the need for training, and their optimistic outlook for future actions. These findings imply no services, support, or systems on the long-term side. The only positively influencing factors that are consistent were the healthcare providers' belief in the problem's existence and their optimistic view.

Many healthcare professionals, including health extension workers, nurses, and physicians, acknowledge that components of severe malaria-related disability occur but do not believe it is their responsibility to do post-malaria screening and provide rehabilitation support for children with severe malaria-related disability. Most relate this factor with the absence of a clear clinical practice guideline in their respective health facilities, while other associate the issue with lack of relevant training, and, as a result, lack of the required knowledge and skills to address the components of disability. Evidence suggests that precise and clear guidance in clinical practice areas improve the quality of malaria case management by healthcare providers [63–65]. Our findings help policymakers and health facilities design relevant mechanisms to incorporate the prevention of severe malaria-related disability and provision of rehabilitation support in the roles and responsibilities of healthcare providers at various levels, by delineating clear scope of practice for each level.

The findings of this study showed that there is, generally, limited knowledge and skills pertinent to addressing severe malaria-related disability. These gaps mainly related to a lack of information about the link between severe malaria and disability. Furthermore, as there has not been a follow-up mechanism for severe malaria survived children, the professionals described that they did not know much about the actual long-term impacts. However, these findings do not reflect the healthcare providers' views regarding the severity of malaria. Nor does it show their beliefs about the potential adverse outcomes of the disease. They all believe that malaria could have severe presentations and have long-term consequences. So, the knowledge and skills gaps were associated with the lack of specific training to provide rehabilitation support for children with severe malaria-related disability. Previous studies also support the role of knowledge/skill in influencing the clinical behavior of healthcare providers [66,67]. Our findings imply a need to incorporate approaches that help address severe malaria-related disability in in-service and pre-service training programs. So, policymakers and other stakeholders working on malaria could benefit from the findings to design relevant training packages that include the application of comprehensive disability frameworks in malaria treatment and care.

In this study, environmental contexts and resources were among the most important factors that negatively influence the prevention of severe malaria-related disability and provision of rehabilitation support. We learned that these factors were multi-dimensional, however, they can broadly be categorized into three: absence of relevant guidelines, scarcity of required resources, and challenging institutional systems and processes. Evidence highlights that healthcare providers' performance is associated with contextual factors such as the local environment (e.g., lack of resources) and staffing (e.g., shortage of health personnel) [67]. As malaria is an infectious disease that can be prevented and has a cure, its clinical approach has been focused on managing acute cases [16]. Our findings show that the healthcare providers have not been provided rehabilitation support for children with severe malaria as part of the national malaria treatment and care protocol. Nor have they attempted to follow-up with children who have survived. According to the participants, this problem was partly linked to a lack of appropriate guidelines or treatment protocols for such cases, and it is because the current national guideline is all about managing acute cases. Evidence supports that using relevant guidelines positively influences healthcare providers' clinical behaviors and improves clinical practice [68]. Future interventions need to include the development of guidelines or protocols that help to provide rehabilitation support for children with severe malaria-related disability.

Institutional systems and processes were identified as negatively influencing factors. In this regard, the most cited issues were a weak patient referral system, and a diversion of health sectors' focus when new epidemics or pandemics emerged. The main gap in the referral system was a lack of feedback about when and in what condition severe malaria survived children were returned to their home or to their community, which created an information gap

regarding the frequency and type of health check-ups. The major problems associated with the emergence of epidemics or pandemics such as the COVID-19 were a cut of budget allocated for malaria-related activities and the diversion of the focus of clinicians and other stakeholders to the new problem [69]. So, these findings will help decision-makers, healthcare providers, and other stakeholders working on malaria to give special attention to health facilities in malaria areas in terms of resource allocation and priority setting.

Another striking finding obtained through analyzing the healthcare providers' descriptions using the TDF was the presence of external social influences that hinder the prevention of severe malaria-related disability and provision of rehabilitation support. These influences include patients' beliefs and socio-cultural factors. Although there is no specifically established approach to address the problems (in the facilities where the participants were working), some of the long-term consequences of severe malaria could have been prevented either by early treatment of severe cases (within 24 hours) or treating some potentially disabling immediate complications (e.g., hypoglycemia, seizure, and so on) as early as possible [17]. However, according to the participants, parents, especially those living in rural areas, do not give much attention to the danger signs and mostly refuse referrals. What worsens the problem is the socio-cultural influences circumscribing some severity signs such as seizures, delirium, and coma. Parents in rural areas, in general, and some people in urban areas associate these issues with "evil eyes" or "evil spirits" and do not take children with those symptoms to a modern health facility [70,71]. Nor do they allow parenteral medications if they bring the children to a health institution. These external influences imply that the beliefs are deep-rooted in the community's belief system and have adversely affected the prevention of severe malaria-related disability. So, future interventions that aim to modify the current clinical practice related to severe malaria need to consider external social influences.

On the other hand, this study identified factors that can positively influence the prevention of severe malaria-related disability and provision of rehabilitation support. The healthcare providers expressed their goals, ambitions, and the things that motivate them in positive statements. Although internal and external factors challenge them, most healthcare providers who participated in the study highlighted their plans to address the problems in the future, hoping that limitations such as skills gap and scarcity of resources will be solved over time. Some also believe that an internal motivation originating from the quick recovery of sick children drives them to carry out their duties most of the time, in line with previous studies [72]. Future implementers will benefit if they design interventions aimed at the goals, ambitions, and intrinsic motivating factors that can potentially modify the perceptions and behaviors of healthcare providers treating children with severe malaria.

Also, the presence of good team spirit, collaboration, and shared understanding of the things they do were additional driving forces described by the healthcare providers involved in this study, in agreement with earlier evidence [73]. Although these facilitating factors were common to nearly all participants, they mentioned them about their respective health institutions. However, their views regarding the relationships between two different health facilities were variable. In some situations, it was regarded positively, but it was described unfavorably in others. These findings imply that the existing optimistic professional relationships, collaboration, and teamwork within a health facility can be used to improve the desired clinical service, whereas interfacility relationships and collaborations are variable and need further evaluation. So, policymakers, researchers, and clinicians could benefit from these findings to design further research or interventions.

## Limitations

Evidence of severe malaria-related disability is still developing [15], and there are currently no guidelines regarding its clinical management. The TDF was designed for use in circumstances where clear evidence-based clinical practice guidelines are in place, which could be a limitation. It has been claimed that the theoretical domains may be less useful in cases where there are no clear guidelines or if the evidence foundation is uncertain as the effect of potential factors may be overwhelmed by differences in attitudes [37,74]. In our situation, this suggestion did not appear to be a problem. Despite differences in practice and attitudes, nine of the fourteen domains were found to be relevant to the phenomenon under investigation. On the other hand, as we recruited healthcare providers only from one region of Northwest Ethiopia with a high malaria prevalence, the findings may be limited in their applicability to other contexts due to regional variations in contextual factors such as infrastructure. However, we took measures to improve this limitation by selecting participants using a purposive sample of healthcare providers from primary health care units, district hospitals, specialized hospitals, and various professional specialties.

## Conclusions

The perceptions and behaviors of healthcare providers who participated in this study almost entirely focused on preventing severe malaria-related disability. Nine factors (theoretical domains) were found to influence the perceptions and behaviors of the healthcare providers: knowledge, skills, beliefs about consequences, goals, optimism, reinforcement, environmental context and resources, social influences, and social or professional role and identity. These findings suggest the need for interventions to support healthcare providers prevent severe malaria-related disability and provide rehabilitation support in their clinical practice. These interventions should target healthcare providers treating children with malaria at all levels of the healthcare system, including primary health care units, district hospitals, and specialized hospitals. The interventions should also focus on developing clinical guidelines to prevent severe malaria-related disability and to provide rehabilitation support for children with severe malaria-related disability, training clinicians on the guidelines, and addressing contextual and resource-related challenges. Additionally, it is crucial to modify external barriers (such as patient beliefs and socio-cultural issues) that negatively impact those preventive and rehabilitation practices. Finally, our findings provide theory-driven baseline evidence (by providing a list of factors) that help determine the predictors of healthcare providers' perceptions and behaviors towards the prevention of severe malaria-related disability and rehabilitation support, which would benefit future research, policy, and practice in multidisciplinary professions, including rehabilitation.

## Supporting information

**S1 Table. Interview guide.**
(DOCX)

**S2 Table. Coding guide.**
(DOCX)

**S3 Table. The proportion of codes addressing prevention.**
(DOCX)

**S4 Table. Consolidated criteria for reporting qualitative studies (COREQ): 32-item checklist.**
(DOC)

## Author Contributions

**Conceptualization:** Eshetu Haileselassie Engeda, Heather M. Aldersey, Colleen M. Davison, Nora Fayed.

**Data curation:** Eshetu Haileselassie Engeda.

**Formal analysis:** Eshetu Haileselassie Engeda.

**Investigation:** Eshetu Haileselassie Engeda.

**Methodology:** Eshetu Haileselassie Engeda, Heather M. Aldersey, Colleen M. Davison, Kassahun Alemu Gelaye, Nora Fayed.

**Project administration:** Eshetu Haileselassie Engeda.

**Resources:** Eshetu Haileselassie Engeda.

**Software:** Eshetu Haileselassie Engeda.

**Supervision:** Nora Fayed.

**Writing – original draft:** Eshetu Haileselassie Engeda.

**Writing – review & editing:** Eshetu Haileselassie Engeda, Heather M. Aldersey, Colleen M. Davison, Kassahun Alemu Gelaye, Nora Fayed.

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
