## [Decision Letter · Decision Letter 0]

26 Sep 2023

PONE-D-23-03146Perceptions and behaviors of healthcare providers towards rehabilitation support to children with severe malaria-related disability in Ethiopia: A qualitative descriptive study using the Theoretical Domains FrameworkPLOS ONE

Dear Dr. Engeda,

Thank you for submitting your manuscript to PLOS ONE. After careful consideration, we feel that it has merit but does not fully meet PLOS ONE’s publication criteria as it currently stands. Therefore, we invite you to submit a revised version of the manuscript that addresses the points raised during the review process.

We look forward to receiving your revised manuscript.

Kind regards,

Taofiki Ajao Sunmonu

Academic Editor

PLOS ONE

Journal Requirements:

2. Please include a complete copy of PLOS’ questionnaire on inclusivity in global research in your revised manuscript. Our policy for research in this area aims to improve transparency in the reporting of research performed outside of researchers’ own country or community. The policy applies to researchers who have travelled to a different country to conduct research, research with Indigenous populations or their lands, and research on cultural artefacts. The questionnaire can also be requested at the journal’s discretion for any other submissions, even if these conditions are not met. Please find more information on the policy and a link to download a blank copy of the questionnaire here: https://journals.plos.org/plosone/s/best-practices-in-research-reporting.  Please upload a completed version of your questionnaire as Supporting Information when you resubmit your manuscript

"We acknowledge the Mastercard Foundation Scholars Program for financially supporting this study. "

"This work was supported by the Mastercard Foundation Scholars Program, which covered personal and material costs as part of a larger research project. Only EE received the grant (grant number not applicable). However, the organization had no role in designing or conducting the study, including data collection, management, analysis, interpretation of the findings, and manuscript writing, review, and approval."

Additional Editor Comments:

The authors need to increase the sample size of participants in the study as siggested by the reviewers . Also various categories of health wiorkers should be be recruited to participate in thestudy. aLSO STANDARD TOOLS AND RESEARCH METHODS SHOULD BE APPLIED IN THE STUDY.

Reviewers' comments:

Reviewer's Responses to Questions

**Comments to the Author**

1. Is the manuscript technically sound, and do the data support the conclusions?

Reviewer #1: Partly

Reviewer #2: Yes

2. Has the statistical analysis been performed appropriately and rigorously? 

Reviewer #1: No

Reviewer #2: Yes

3. Have the authors made all data underlying the findings in their manuscript fully available?

Reviewer #1: Yes

Reviewer #2: Yes

4. Is the manuscript presented in an intelligible fashion and written in standard English?

Reviewer #1: Yes

Reviewer #2: Yes

5. Review Comments to the Author

Reviewer #1: As the sampling frame and sampling units have wide variation as well as sample size is too small.

Under these circumstances, there is every possibility that the inferences drawn and conclusions made may be having bias.

It is suggested that sample size may be increased so that each category of respondents are adequately represented. Possibility of subgroup analysis may also be attempted.

Reviewer #2: Paper is well written. Some aspects to clarify/address:

1. Provide a clear/standard definition for severe malaria related disability in the context of this study -What is the scope? As mentioned on page 3 line 60 to 63 most published literature on malaria related complications are discussed in acute context . Is this disability limited to neuro-cognitive disability outside of acute complication of malaria? On page 3 lines 65 and 66 mentions known rehabilitation and interventions have focused on neurocognitive impairment.

2. Was there prior consideration of other frameworks such as CFIR (widely used in low and middle income countries) before the selection of TDF in this context where there is no clear evidence based guideline/protocol for management of severe malaria related disability? Was any consideration that TDF and CFIR approach include determinants at both individual and collective level? Would combining the CFIR and TDF help to more fully define the multi-level nature of behavior change in healthcare organizations be a better aproach than using either of them alone?

3. For sampling-how was the number of 13 health workers arrived at? Representation of different cadres mentioned; did any health workers decline to participate and if so why?

4. In the section on “Trustworthiness” Comment on positionality of EE having established relationship with the participants over time and his involvement in analysis-would this introduce any bias and how was this minimized?

5. In the analysis approach after codebook development and pilot by EE and colleague-was this followed by primary coding and secondary coding by different researchers? What was the approach to the resolution of queries or disagreement in coding-Were all researchers involved or some of them?

6. Was any consideration given to health workers caring for children with preexisting conditions who may suffer more severe disability from malaria for example children with anemia due to differnt causes example sickle cell disease –prevalent in some East African countries like Kenya Uganda and Tanzania…Is this prevalent in Ethiopia?

6. PLOS authors have the option to publish the peer review history of their article (what does this mean?). If published, this will include your full peer review and any attached files.

Reviewer #1: No

Reviewer #2: **Yes: **Ednah Akinyi Ojee

---

## [Author Response · Author response to Decision Letter 0]

15 Nov 2023

Responses to reviewer #1

Comments/questions: As the sampling frame and sampling units have wide variation as well as sample size is too small.

Under these circumstances, there is every possibility that the inferences drawn, and conclusions made may be having bias.

It is suggested that sample size may be increased so that each category of respondents are adequately represented. Possibility of subgroup analysis may also be attempted.

Responses: Thank you so much for your valuable comments, and we understand your concerns. In contrast to research that applies theoretical data saturation, such as the grounded theory, this qualitative descriptive study did not attempt to fully describe all aspects regarding the perceptions and behaviors of healthcare providers, nor does the study aim to identify either consistency or substantial differences in the perceptions and behaviors of the various professionals included in the study, which could lead to a sub-group analysis. Instead, we focused on a comprehensive understanding of the phenomenon and identifying communalities. Including various professionals helped us enriched the data from different perspectives and contexts. 

Therefore, participant recruitment was stopped after collecting a rich and varied account of the phenomenon under investigation by applying the concept of "information power" and the techniques recommended by Francis and colleagues (2010) for theory-informed qualitative studies, cited in the manuscript. No new information was obtained when three participants were interviewed following a preliminary analysis of the data obtained from the first 10 participants; therefore, continuing the data collection after the 13th interview was unnecessary.

Responses to reviewer #2

Comment/question #1: Provide a clear/standard definition for severe malaria related disability in the context of this study -What is the scope? As mentioned on page 3 line 60 to 63 most published literature on malaria related complications are discussed in acute context. Is this disability limited to neuro-cognitive disability outside of acute complication of malaria? On page 3 lines 65 and 66 mentions known rehabilitation and interventions have focused on neurocognitive impairment.

Response: Thank you so much for your valuable feedback. 

Severe malaria is a laboratory-confirmed malaria characterized by one or more clinical symptoms or laboratory findings, including impaired consciousness, respiratory distress, multiple convulsions, prostration, abnormal bleeding, jaundice, hypoglycemia, acidosis, hyperlactatemia, renal impairment, hyperparasitemia, cerebral malaria, and severe malarial anemia. In line with the World Health Organization's standard definition of disability, any form of impairment (e.g., attention deficits), activity limitation (e.g., behavioral problems), and participation restriction (e.g., negative peer attitudes) that occurred because of this health condition was considered as severe malaria-related disability. Therefore, neurocognitive impairments are included but are not the only forms of severe malaria-related disability. 

We included these standard definitions in the revised manuscript (page 3 line 50-56) in track changes. 

Comment/question #2: Was there prior consideration of other frameworks such as CFIR (widely used in low- and middle-income countries) before the selection of TDF in this context where there is no clear evidence based guideline/protocol for management of severe malaria related disability? Was any consideration that TDF and CFIR approach include determinants at both individual and collective level? Would combining the CFIR and TDF help to more fully define the multi-level nature of behavior change in healthcare organizations be a better approach than using either of them alone?

Response: We compared other determinant frameworks, such as the active implementation framework, CIFIR, and PARIHS, before deciding on the TDF. We preferred TDF to other frameworks because our primary goal was to identify key theoretical constructs pertinent to healthcare providers' perceptions and behaviors towards rehabilitation support to children with severe malaria-related disability so that these constructs will be accessible for future large-scale quantitative studies and help develop strategies for effective implementation. 

We agree that CFIR is a widely used framework in low- and middle-income countries and is among the top five most accessed articles in implementation science. However, we believe that, for the objective we formulated, our selection of the TDF was also appropriate, given that the TDF contains 33 behavior change theories that are accessible in 14 theoretical domains, which can capture healthcare providers' perceptions, behaviors, and (unlike most other behavioral theories) environmental and institutional factors. 

We also agree that combining CIFR and TDF could help us more fully understand multi-level behavior change than using individual frameworks. However, we did not aim to fully describe all aspects of behavior change in our qualitative descriptive study. Instead, we focused on a comprehensive understanding of the phenomenon. We will consider combining the two frameworks in our future large-scale studies. 

Comment/question #3: For sampling-how was the number of 13 health workers arrived at? Representation of different cadres mentioned; did any health workers decline to participate and if so why?

Response: Participant recruitment was stopped after collecting a rich and varied account of the phenomenon under investigation by applying the concept of "information power" and the techniques recommended by Francis and colleagues (2010) for theory-informed qualitative studies, cited in the manuscript. No new information was obtained when three participants were interviewed following a preliminary analysis of the data obtained from the first 10 participants; therefore, continuing the data collection after the 13th interview was unnecessary.

We used a purposive sampling technique based on criteria, and no participant declined recruitment or withdrew from participation after recruitment.

Comment/question #4: In the section on “Trustworthiness” Comment on positionality of EE having established relationship with the participants over time and his involvement in analysis-would this introduce any bias and how was this minimized?

Response: We used various methods to ensure trustworthiness, including prolonged engagement, audit trail, debriefing, member check, and reflexivity. Establishing relationships was part of the prolonged engagement and was necessary. However, we continuously reflected on the process and our feelings to minimize bias. As an interviewer, I paid particular attention to this aspect and documented a reflexivity statement during the data collection period. The reflexivity continued throughout the research process, including data analysis. To that end, the prolonged engagement contributed to obtaining rich data. It did not introduce significant bias to the extent that affected the findings, as we were aware of our actions and feelings through continuous reflexive journaling. However, we are not arguing that we avoided bias 100%, as complete bracketing is impossible. 

We described this in the manuscript on page 10, lines 193-195, by adding a statement in track changes.

Comment/question #5: In the analysis approach after codebook development and pilot by EE and colleague-was this followed by primary coding and secondary coding by different researchers? What was the approach to the resolution of queries or disagreement in coding-Were all researchers involved or some of them?

Response: Primary and secondary coding was done by EE and NF using a coding guide, and the reliability of coding was checked first by proportion and then by Kappa statistics. The other team members were also involved in a discussion to resolve a few codes that the two coders disagreed on, which helped finalize the coding process. 

We provided this description in the manuscript, page 8, lines 176-179 in track changes.

Comment/question #6: Was any consideration given to health workers caring for children with preexisting conditions who may suffer more severe disability from malaria for example children with anemia due to different causes example sickle cell disease –prevalent in some East African countries like Kenya Uganda and Tanzania…Is this prevalent in Ethiopia?

Response: In our study, severe malaria was defined as laboratory-confirmed malaria characterized by one or more clinical symptoms or laboratory findings, including impaired consciousness, respiratory distress, multiple convulsions, prostration, abnormal bleeding, jaundice, hypoglycemia, acidosis, hyperlactatemia, renal impairment, hyperparasitemia, cerebral malaria, and severe malarial anemia. Therefore, disability resulting from other con-existing conditions was not considered. 

We did not study the prevalence of severe malaria-related disability; however, as malaria is prevalent in nearly 60% of the country (Ethiopia), a substantial number of children with severe malaria-related disability is expected. Our recently published qualitative study also gives clues for that (Severe malaria-related disability in Ethiopian children from the perspectives of caregivers: an interpretive description study - PubMed (nih.gov)). 

Responses to additional editor comments

Editor comments: The authors need to increase the sample size of participants in the study as suggested by the reviewers. Also, various categories of health workers should be recruited to participate in the study. Also, STANDARD TOOLS AND RESEARCH METHODS SHOULD BE APPLIED IN THE STUDY

Response: Dear the editor, thank you so much for these additional comments. 

Regarding the sample size, we believe that we had adequate sample size, given the purpose of our study and the methods we applied including the concept of ‘information power’ (Sample Size in Qualitative Interview Studies: Guided by Information Power - Kirsti Malterud, Volkert Dirk Siersma, Ann Dorrit Guassora, 2016 (sagepub.com)). Moreover, we applied Francis and colleagues’ (2010) recommendations to determine sample size for a theory-informed qualitative descriptive study (https://pubmed.ncbi.nlm.nih.gov/20204937/). Also, as a background reading, we reviewed other widely discussed ideas and recommendation around the concept of data saturation in qualitative research such as the Braun and Clerck’s argument (To saturate or not to saturate? Questioning data saturation as a useful concept for thematic analysis and sample-size rationales: Qualitative Research in Sport, Exercise and Health: Vol 13, No 2 (tandfonline.com).

Unlike quantitative designs, the qualitative approaches do not have straightforward rules of sample size calculation. Instead, they depend on data adequacy guided by certain principles such as ‘saturation’ and ‘information power’. In our case, as we used purposive sampling to select ‘information rich’ participants (healthcare providers who told their experiences in a well articulated manner, we collected rich or adequate data by interviewing 13 participants. A recent systematic review that assessed saturation in qualitative studies shows that 9-17 interviews reached saturation (Sample sizes for saturation in qualitative research: A systematic review of empirical tests - ScienceDirect). 

On the other hand, this qualitative descriptive study focuses on a comprehensive understanding of healthcare providers' perceptions and behaviors without attempting to identify consistency or differences among each group of healthcare providers. Instead, it identifies commonalities and enriches data from different perspectives and contexts. Therefore, sub-group analysis was not our aim.

Regarding the recommendation on the application of standard tools and methods, we are confident that we applied all the requirements for a qualitative descriptive study and applied the Consolidated criteria for reporting qualitative studies (COREQ) (Consolidated criteria for reporting qualitative research (COREQ): a 32-item checklist for interviews and focus groups | EQUATOR Network (equator-network.org)). Unlike quantitative studies, qualitative studies do not utilize standardized tools or instruments for data collection. Therefore, we did not use a standardized tool; instead, we used an interview guide developed by the research team and refined after pilot interviews. We used a published method article for data analysis and followed all the steps recommended. Our methodology and methods are explicitly described on pages 5-10 of the manuscript. 

One of the four questions presented to the reviewers does not apply to our study design. For the question “Has the statistical analysis been performed appropriately and rigorously?” reviewer #1 said “No,” whereas reviewer #2 said “Yes.” However, statistical analysis is not applicable in our qualitative study.

---

## [Decision Letter · Decision Letter 1]

22 Jan 2024

PONE-D-23-03146R1Perceptions and behaviors of healthcare providers towards rehabilitation support to children with severe malaria-related disability in Ethiopia: A qualitative descriptive study using the Theoretical Domains FrameworkPLOS ONE

Dear Dr. Engeda,

Thank you for submitting your manuscript to PLOS ONE. After careful consideration, we feel that it has merit but does not fully meet PLOS ONE’s publication criteria as it currently stands. Therefore, we invite you to submit a revised version of the manuscript that addresses the points raised during the review process. 

We look forward to receiving your revised manuscript.

Kind regards,

Edison Arwanire Mworozi, M.D

Academic Editor

PLOS ONE

Journal Requirements:

**Additional Editor Comments:**

The manuscript has been reviewed and as recommended by two reviewers, please revise accordingly.

Reviewers' comments:

Reviewer's Responses to Questions

**Comments to the Author**

1. If the authors have adequately addressed your comments raised in a previous round of review and you feel that this manuscript is now acceptable for publication, you may indicate that here to bypass the “Comments to the Author” section, enter your conflict of interest statement in the “Confidential to Editor” section, and submit your "Accept" recommendation.

Reviewer #2: All comments have been addressed

Reviewer #3: (No Response)

Reviewer #4: All comments have been addressed

Reviewer #5: (No Response)

2. Is the manuscript technically sound, and do the data support the conclusions?

Reviewer #2: Yes

Reviewer #3: Yes

Reviewer #4: Yes

Reviewer #5: Yes

3. Has the statistical analysis been performed appropriately and rigorously? 

Reviewer #2: N/A

Reviewer #3: Yes

Reviewer #4: Yes

Reviewer #5: N/A

4. Have the authors made all data underlying the findings in their manuscript fully available?

Reviewer #2: Yes

Reviewer #3: Yes

Reviewer #4: Yes

Reviewer #5: Yes

5. Is the manuscript presented in an intelligible fashion and written in standard English?

Reviewer #2: Yes

Reviewer #3: Yes

Reviewer #4: Yes

Reviewer #5: Yes

6. Review Comments to the Author

Reviewer #2: The authors have responded adequately to all my questions and I am satisfied with this revision.

Regarding their comment on the question that asks whether statistical analysis has been performed appropriately and rigorously where reviewer 1 responded “No” and reviewer 2 “Yes”

As the 2nd reviewer, my initial response “Yes” to this was based on interpretation of this question in the context of a qualitative study to be asking whether the qualitative analysis had been performed appropriately to acceptable standards for reproducibility of this research work.

Reviewer #3: One minor suggestion in line 163:non-verbal expressions of participants, and his feelings (50). I suggest that you delete "his"

Reviewer #4: Dear Authors,

Thank you for submitting your well-written manuscript for review. I appreciate your careful research and the insights you've provided. However, I have some suggestions, questions, and corrections that I believe will further strengthen your manuscript.

Abbreviations: Please check the consistency of abbreviations throughout the manuscript, especially at L54. Ensure that all abbreviations are defined clearly at their first mention.

ICF Abbreviation: On page 62, please use the correct abbreviation.

Citations: The current citation style needs to be refined to adhere to the journal's guidelines. Please review the citation style guide and ensure that all references are cited correctly.

Contributions: While I understand that the EE played a significant role in the practical aspects of the study, I recommend adding a contribution statement at the end of the manuscript. This will provide a clear overview of each author's contribution to the research.

Reference Clarification: At L163, please provide more specific information about the reference you've cited.

VerSatim Software: At L164, please include the company name and country of origin for the Versatim software mentioned.

NVivo Software: At L167, please add more context about the NVivo software, such as its purpose and applications.

Amhanic: Please explicitly mention the language after the first time Amhanic is used in the manuscript, ensuring clarity for readers.

Health Care Selection: I'm curious about the basis for selecting the health care facilities you've included in your study. Please provide more details about your selection criteria.

Health Care Availability: Can you provide information about the total number of health care facilities in the country and explain why you chose to focus on these specific ones? Are they representative of the overall health care landscape?

I believe that addressing these suggestions, questions, and corrections will further enhance the manuscript's clarity, rigor, and overall quality. Thank you for your attention to these details.

Reviewer #5: Minor grammatical and clarification of some sentences needed. Further comments are included in the article which has been uploaded.

7. PLOS authors have the option to publish the peer review history of their article (what does this mean?). If published, this will include your full peer review and any attached files.

Reviewer #2: **Yes: **Ednah Ojee

Reviewer #3: **Yes: **Ahmed Adeel

Reviewer #4: No

Reviewer #5: **Yes: **Dr. Alberta Amu

---

## [Author Response · Author response to Decision Letter 1]

24 Jan 2024

Response to Reviewer #2

Comments/questions:

The authors have responded adequately to all my questions, and I am satisfied with this revision.

Response: Thank you so much for your time and scholarly feedback; your comments helped us improve our manuscript. 

Regarding their comment on the question that asks whether statistical analysis has been performed appropriately and rigorously where reviewer 1 responded “No” and reviewer 2 “Yes”

As the 2nd reviewer, my initial response “Yes” to this was based on interpretation of this question in the context of a qualitative study to be asking whether the qualitative analysis had been performed appropriately to acceptable standards for reproducibility of this research work.

Response: Yes, that makes sense. 

Responses to Reviewer #3

One minor suggestion in line 163: non-verbal expressions of participants, and his feelings (50). I suggest that you delete "his"

Response: Thank you so much for your time and comment. The pronoun ‘his’ has been deleted. 

Responses to Reviewer #4

Thank you for submitting your well-written manuscript for review. I appreciate your careful research and the insights you've provided. However, I have some suggestions, questions, and corrections that I believe will further strengthen your manuscript.

Response: Thank you so much for your rigorous review and detailed comments. 

Abbreviations: Please check the consistency of abbreviations throughout the manuscript, especially at L54. Ensure that all abbreviations are defined clearly at their first mention.

Response: We have re-checked the abbreviations throughout the document. We used abbreviations when a text that needs to be abbreviated appears more than three times. In line 54, the World Health Organization appears only twice, and we prefer to spell it out instead of using the abbreviation WHO. 

ICF Abbreviation: On page 62, please use the correct abbreviation

Response: The standard abbreviation for the ‘International Classification of Functioning Disability and Health’ is ICF, as abbreviated by the World Health Organization (WHO). Please see in the following link: International Classification of Functioning, Disability and Health (ICF) (who.int)

Citations: The current citation style needs to be refined to adhere to the journal's guidelines. Please review the citation style guide and ensure that all references are cited correctly.

Response: We used the Vancouver referencing style as highlighted in PLOS ONE’s authors’ guideline: “PLOS uses the reference style outlined by the International Committee of Medical Journal Editors (ICMJE), also referred to as the “Vancouver” style.” As we also used the EndNote citation manages, we believe that the references have been inserted correctly. 

Contributions: While I understand that the EE played a significant role in the practical aspects of the study, I recommend adding a contribution statement at the end of the manuscript. This will provide a clear overview of each author's contribution to the research.

Response: A contribution statement has been added at the end of the manuscript. Please see the tracked changes (L776). 

Reference Clarification: At L163, please provide more specific information about the reference you've cited.

Response: The following clarification is added ‘as recommended by Phillipi and Lauderdale (2018).

Verbatim Software: At L164, please include the company name and country of origin for the Verbatim software mentioned.

Response: We did not use software for the verbatim transcription; instead, research assistants transcribed it. This clarification is added to the document. 

NVivo Software: At L167, please add more context about the NVivo software, such as its purpose and applications.

Response: We re-phrased the section through L167-169 to address your comment: “We imported the transcripts into NVivo 12 Plus software to facilitate the data analysis. The purpose of the software was to assist the researchers in organizing, visualizing, storing, and reporting their data; otherwise, the entire analysis was a cognitive process.”

Amharic: Please explicitly mention the language after the first time Amharic is used in the manuscript, ensuring clarity for readers.

Response: We added the statement described below (with references) to clarify how we used Amharic (the local language). 

“To minimize loss of meaning during translation, we analyzed the data using Amharic (the local language) until we generated preliminary themes. After that, the codes and associated data were translated into English.” (L170-172)

Health Care Selection: I'm curious about the basis for selecting the health care facilities you've included in your study. Please provide more details about your selection criteria.

Response: In line with the nature of the qualitative design, we selected both the setting and participants purposively with criteria. The criteria for selecting the healthcare facilities were (1) their level (primary, secondary, and tertiary), and (2) healthcare facilities found in malaria endemic areas. These criteria are included in the revised version (L138-139).

Health Care Availability: Can you provide information about the total number of health care facilities in the country and explain why you chose to focus on these specific ones? Are they representative of the overall health care landscape?

Response: Unlike quantitative studies, which commonly aim for generalization, the qualitative approach does not aim for that, as generalizing the findings is not its purpose. Instead, it aims for transferability, defined as “The degree to which the results of qualitative research can be transferred to other contexts or settings with other respondents. The researcher facilitates the transferability judgment by a potential user through thick description.” (Series: Practical guidance to qualitative research. Part 4: Trustworthiness and publishing (tandfonline.com); p. 121). Collecting ‘representative’ samples is associated with generalizability and sounds like a post-positivist axiology, which differs from our naturalistic perspective in which collecting a representative sample is less valued. Instead, we choose samples purposively and keep recruiting participants until data and meaning saturation happens and adequacy of information is ensured.

Therefore, in line with our philosophical worldview, we did not consider representativeness and generalizability but transferability. What is critical to ensure transferability, in addition to the behaviors of the healthcare providers, is the context related to the phenomenon under investigation. Therefore, we added a detailed description of the context of the Ethiopian healthcare facilities in relation to malaria treatment and care. Please see L140 of the revised version in the tracked changes.

I believe that addressing these suggestions, questions, and corrections will further enhance the manuscript's clarity, rigor, and overall quality. Thank you for your attention to these details.

Response: Yes, your comments are indispensable and help us improve our manuscript. Thank you so much. 

Responses to Reviewer #5

Minor grammatical and clarification of some sentences needed. Further comments are included in the article which has been uploaded.

Response: Thank you so much for your comments. We have corrected the grammatical issues and clarified some sentences in the document per your suggestions. The changes we made are described below and in tracked changes in the manuscript.

L230 Table 2: General Practitioners; qualify if they are Physician General Practitioners

Response: Yes, they are Physician General Practitioners, and it is corrected in the document. 

L238 “We also delineated the proportion of codes addressed the prevention of severe malaria-related disability” Clarify this

Response: Clarification added, and the modified statement is as follows: “We also delineated the proportion of codes that addressed the prevention of severe malaria-related disability, indicating acute care of severe malaria instead of rehabilitation support and care.”

L396 “On the other side of the spectrum, many participants described that the focus of the health facilities could be diverted when new public health important disease emerge, and this factors negatively influence the amount of budget allocated for malaria-related activities.” re-phrase; grammatical error 

Response: It is corrected, and the sentence has been re-phrased as follows: “On the other side of the spectrum, many participants described that the focus of the health facilities could be diverted when new pandemics or epidemics emerge, and these factors negatively influence the budget allocated for malaria-related activities.”

L506-507 “Our findings show that the healthcare providers 24 507 have not been systematically provided rehabilitation support for children with severe malaria.” Clarify this sentence. 

Response: The sentence is re-phrased for clarification as follows:

“Our findings show that the healthcare providers have not been provided rehabilitation support for children with severe malaria as part of the national malaria treatment and care protocol.”

L516-518 “The main gap in the referral system was a lack of feedback about when and on what condition severe malaria survived children were returned to their home or to their community, which created an information gap regarding the frequency and type of health check-ups.” In the phrase “when and on what condition” in or on?

Response: It should be ‘in.’ Corrected in the document. 

L533 “What worsens the problem is the socio-cultural influences circumscribing some severity signs such as seizures, delirium, and coma.” How do the sociocultural issues restrict some severity signs?

Response: It is explained in the next sentence in the document (L533-534), highlighting that “Parents in rural areas, in general, and some people in urban areas associate these issues with “evil eyes” or “evil spirits” and do not take children with those symptoms to a modern health facility.”

L540 “On the contrary …” to what?

Response: The phrase “On the contrary” is replaced by “On the other hand.”

---

## [Editor Report · Decision Letter 2]

31 Jan 2024

Perceptions and behaviors of healthcare providers towards rehabilitation support to children with severe malaria-related disability in Ethiopia: A qualitative descriptive study using the Theoretical Domains Framework

PONE-D-23-03146R2

Dear Dr. Engeda,

We’re pleased to inform you that your manuscript has been judged scientifically suitable for publication and will be formally accepted for publication once it meets all outstanding technical requirements.

Kind regards,

Edison Arwanire Mworozi, M.D

Academic Editor

PLOS ONE